# ESCRTs function directly on the lysosome membrane to downregulate ubiquitinated lysosomal membrane proteins

Lu Zhu[1,2†], Jeff R Jorgensen[1,2†], Ming Li[1,3‡], Ya-Shan Chuang[1,2], Scott D Emr[1]*

[1]Weill Institute for Cell and Molecular Biology, Cornell University, Ithaca, United States; [2]Department of Molecular Biology and Genetics, Cornell University, Ithaca, United States; [3]Department of Molecular, Cellular and Developmental Biology, University of Michigan, Ann Arbor, United States

**Abstract** The lysosome plays an important role in maintaining cellular nutrient homeostasis. Regulation of nutrient storage can occur by the ubiquitination of certain transporters that are then sorted into the lysosome lumen for degradation. To better understand the underlying mechanism of this process, we performed genetic screens to identify components of the sorting machinery required for vacuole membrane protein degradation. These screens uncovered genes that encode a ubiquitin ligase complex, components of the PtdIns 3-kinase complex, and the ESCRT machinery. We developed a novel ubiquitination system, *Ra*pamycin-*In*duced *Deg*radation (RapiDeg), to test the sorting defects caused by these mutants. These tests revealed that ubiquitinated vacuole membrane proteins recruit ESCRTs to the vacuole surface, where they mediate cargo sorting and direct cargo delivery into the vacuole lumen. Our findings demonstrate that the ESCRTs can function at both the late endosome and the vacuole membrane to mediate cargo sorting and intra-luminal vesicle formation.

*For correspondence: sde26@cornell.edu

†These authors contributed equally to this work

Present address: ‡Department of Molecular, Cellular and Developmental Biology, University of Michigan, Ann Arbor, United States

Competing interests: The authors declare that no competing interests exist.

## Introduction

The lysosome is best known as a degradative organelle, but it also plays an important role in maintaining ion and nutrient homeostasis in the cell. Similar to the plasma membrane (PM), the lysosomal membrane contains various transporters that mediate the efflux and influx of ions and nutrients across the lysosomal membrane (*Xu and Ren, 2015*). Dysregulation of lysosomal transporters underlies the pathogenesis of many inherited metabolic disorders in humans termed lysosomal storage diseases (LSDs), such as Niemann-Pick disease, Salla disease, Cystinosis, and mucolipidosis (*Gahl et al., 2002*; *Parkinson-Lawrence et al., 2010*; *Schulze and Sandhoff, 2011*; *Vitner et al., 2010*). The precise regulation of lysosomal membrane proteins is therefore crucial to maintain nutrient and ion balance in the cell.

Lysosomes serve as the terminal destination for the delivery of cellular proteins targeted for degradation, but how lysosomes turnover their own resident membrane proteins remains poorly understood. Using the yeast *Saccharomyces cerevisiae*, our lab recently found that resident transporters on the vacuole membrane (analogous to the lysosomal membrane in higher eukaryotes) are sorted to the vacuole lumen and degraded upon specific stimuli. Specifically, we found that the vacuole membrane lysine transporter Ypq1 is ubiquitinated after lysine withdrawal by Rsp5, a Nedd4 family HECT-type E3 ubiquitin ligase. Rsp5 is recruited to the vacuole membrane by Ssh4, a PY-motif containing type-I transmembrane protein. After ubiquitination, Ypq1 is sorted into the vacuole lumen for degradation (*Li et al., 2015b*). We also reported that the zinc transporter Cot1 is ubiquitinated by Tul1, a RING-type E3 ligase, and is sorted into the vacuole lumen for degradation upon the

depletion of zinc (*Li et al., 2015a*). Despite using different ubiquitination complexes, these cargoes likely use the same downstream sorting machineries. The trafficking pathway for the degradation of these vacuolar membrane proteins, however, has not yet been elucidated.

Our lab proposed two models for the sorting of ubiquitinated Ypq1. In the first model, Ypq1 is recycled from the vacuole membrane to an intermediate compartment (the late endosome) where the ESCRTs would sort it into intraluminal vesicles (ILVs) before delivery into the vacuole lumen. In the second model, Ypq1 is internalized directly into the vacuole lumen by the ESCRT machinery. Initially, we favored the recycling model (*Li et al., 2015a*, *Li et al., 2015b*) because some Ypq1-GFP puncta detached from vacuole membrane were detected after lysine withdrawal. The recycling model was also consistent with the previously characterized functional location of the ESCRT complexes at the late endosome (*Babst et al., 2002*, *2000*; *Cheever et al., 2001*; *Henne et al., 2012*; *Katzmann et al., 2003*), and ESCRT function directly on the vacuole membrane has never been reported.

To better understand the sorting and trafficking steps for Ypq1 degradation, we performed a genetic selection as well as a genome-wide screen for mutants defective for Ypq1 degradation. Through multiple genetic approaches, we hoped to reveal genes and trafficking pathways required for Ypq1 sorting and degradation. Here, we demonstrate that endosome-vacuole fusion is not required for Ypq1 sorting and degradation, which is inconsistent with the recycling model. Instead, we show that ESCRT complexes function directly on the vacuole surface to facilitate Ypq1 degradation, extending the role for ESCRTs in yeast.

## Results

### Genetic selection for mutants defective in vacuole membrane protein degradation pathway

We have previously shown that Ypq1 sorting from the vacuole membrane into the vacuole lumen after lysine withdrawal is ubiquitin dependent. However, the machinery required for downstream Ypq1 sorting steps remained elusive. To better understand the vacuole membrane protein sorting pathway, we developed genetic approaches to identify new machinery required for this process.

Serendipitously, we found that the insertion of a 38 amino acid streptavidin-binding-peptide (SBP) (*Terpe, 2003*) between Ypq1 and GFP (Ypq1-SBP-GFP) results in constitutive sorting of Ypq1 into the vacuole lumen, even in the presence of lysine (*Figure 1A*). Importantly, Ypq1-SBP-GFP sorting is blocked on the vacuole membrane in a mutant lacking the vacuolar ubiquitin E3 ligase adaptor Ssh4(*Figure 1B*), suggesting that Ypq1-SBP-GFP is degraded through the same vacuole membrane protein degradation pathway as Ypq1-GFP. This new fusion protein allowed us to design a robust screening platform to reveal cellular pathways necessary for Ypq1 sorting.

To design a genetic selection for mutants missorting Ypq1, we built a Ypq1-SBP-GFP-His3 reporter protein, which is constitutively sorted into the vacuole lumen and fails to restore histidine prototrophy in *his3* mutant cells (*Figure 1C*, left). In contrast, mutants defective for Ypq1 ubiquitination and sorting (*Figure 1C*, right), such as *ssh4Δ*, are His+ (*Figure 1D*) because the fused His3 persists on the cytoplasmic surface of the vacuole.

We also designed a genome-wide fluorescence-based screen to identify Ypq1 sorting mutants. To accomplish this, we fused Ypq1-SBP to pHluorin, a pH-sensitive GFP variant that is quenched at low pH (*Miesenböck et al., 1998*; *Prosser et al., 2010*). Since the vacuole lumen is acidic, Ypq1-SBP-pHluorin is quenched once sorted into the vacuole lumen (*Figure 1—figure supplement 1A*). This makes Ypq1 sorting mutants fluorescent, such as *ssh4Δ* (*Figure 1—figure supplement 1B*), whereas WT cells are quenched. We expressed Ypq1-SBP-pHluorin in each of the ~4600 yeast non-essential knock-out collection mutants and quantified their fluorescence using flow cytometry (*Supplemental file 1* and *Figure 1—figure supplement 1C,D*), then validated the hits using light microscopy.

Using these complementary genetic approaches, as shown in the *Supplemental file 2*, we identified the ESCRTs, the E3 ligase *RSP5* and its vacuole membrane adaptor *SSH4*, and the endosome-vacuole fusion machinery(HOPS/CORVET/SNAREs). We also found the PtdIns 3-kinase complex components *VPS34* and *VPS15*, which are essential for making PtdIns(3)P, a prerequisite for ESCRT recruitment to membranes (*Katzmann et al., 2003*). Unexpectedly, despite finding the vacuole

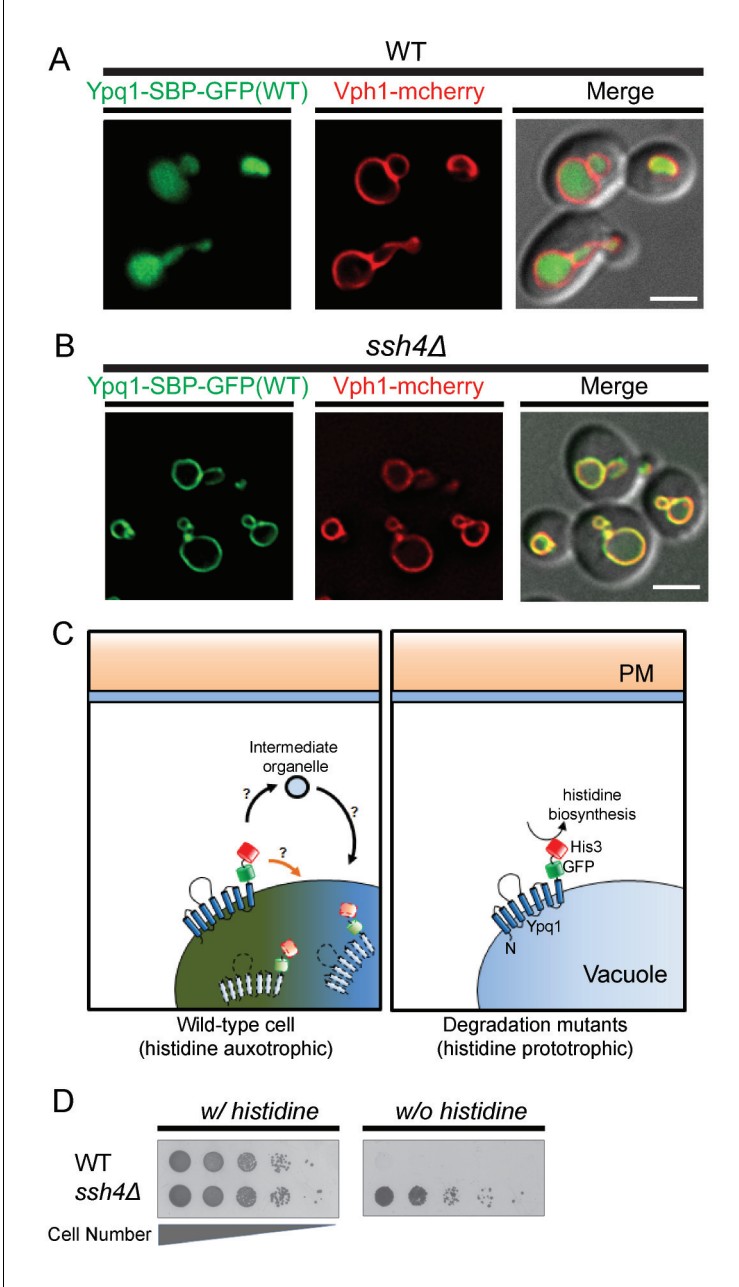

**Figure 1.** Genetic selection for mutations that block the Ypq1 sorting pathway. (**A**) Fluorescent microscopy analysis of cells expressing Ypq1-SBP-GFP and vacuole membrane marker Vph1-mCherry in wild-type and (**B**) *ssh4Δ* cells grown to mid-log phase at 30°C. (**C**) Cartoon depicting genetic selection: wild-type cells constitutively degrade Ypq1-SBP-GFP-His3 via the vacuole membrane protein degradation pathway. This effectively removes His3 from the cytoplasm, making WT cells His-. Degradation mutants, however, prevent Ypq1 sorting, this leaves His3 exposed to the cytoplasm, making them His+. (**D**) Cell growth assay comparing wild-type and *ssh4Δ* mutant cells grown in the presence, or absence of histidine. Scale bars represent 2 μm.

The following figure supplement is available for figure 1:

**Figure supplement 1.** Ypq1-SBP-pHluorin visual screen.

fusion machinery, we did not identify any vesicle coat proteins, vesicle budding machinery or ubiquitin-binding proteins (other than the ESCRT proteins).

## ESCRT and endosome-vacuole fusion mutants block Ypq1 ubiquitination

Since many of the Ypq1 sorting mutants identified in the screens also have established roles in the vacuolar protein sorting (VPS) pathway, we decided to test if these mutants affect Ypq1 ubiquitination by blocking the delivery of Ssh4 to the vacuole membrane. As shown in *Figure 2A*, Ssh4-mNeonGreen is mainly localized to the vacuole membrane and lumen in WT cells. In the ESCRT mutant *vps27Δ*, however, Ssh4 is blocked at aberrant endosomes and co-localizes with the v-ATPase component Vph1, which is trafficked to the vacuole via the VPS pathway and accumulates at aberrant endosomes in ESCRT mutants (*Gerrard et al., 2000*; *Piper et al., 1995*). These data demonstrate that Ssh4 requires ESCRT function to be delivered to the vacuole, which implies that ESCRTs are required for Ypq1 ubiquitination.

To examine this ESCRT-dependence more directly, we monitored the ubiquitination status of Ypq1 after lysine withdrawal in ESCRT mutants (see 'Materials and methods'). We co-expressed Ypq1-GFP and myc-Ub in *doa4Δ* cells. Doa4 is a major deubiquitinase of the endomembrane system, and its deletion stabilizes ubiquitinated forms of Ypq1 (*Amerik et al., 2000*; *Li et al., 2015b*). We immunoprecipitated Ypq1-GFP before and after lysine withdrawal, and probed the precipitates for myc-Ub. In WT cells after 2 hr lysine withdrawal, there was a high-molecular-weight myc-ub signal, indicating that after lysine withdrawal Ypq1 becomes polyubiquitinated. In *vps27Δ*, however, this polyubiquitination did not occur. Taken together, these data demonstrate that in ESCRT mutants, Ypq1 cannot be ubiquitinated because Ssh4 delivery to the vacuole is blocked (*Figure 2B*).

We previously observed that Ypq1 is blocked at the vacuole membrane after lysine withdrawal in the endosome-vacuole fusion mutants *vam3ts* (SNARE) and *vps18ts* (HOPS) at non-permissive temperature (*Li et al., 2015b*). Since Ssh4 delivery requires these fusion machineries (*Figure 2—figure supplement 1A*), we hypothesized that the delivery of newly synthesized Ssh4 may be required and that the pre-existing vacuole membrane pool of Ssh4 may not be sufficient to ubiquitinate Ypq1 after temperature shift. We then tested whether these mutants block Ypq1 ubiquitination. As can be seen in the *Figure 2—figure supplement 1B*, these mutants block the ubiquitination of Ypq1-GFP after lysine withdrawal.

Taken together, our results suggest that proper delivery of Ssh4 to the vacuole membrane is required for the normal ubiquitination of Ypq1-GFP.

## Rapamycin-induced degradation (RapiDeg) system

The Ypq1 ubiquitination defect in ESCRT mutants and endosome-vacuole fusion mutants make it difficult to study whether these components are involved in the downstream sorting steps of ubiquitinated Ypq1. To circumvent this problem, we developed a rapid and specific cargo ubiquitination system, *Rap*amycin-*I*nduced *Deg*radation (RapiDeg), to bypass the normal Ssh4/Rsp5 ubiquitination apparatus by directly targeting ubiquitin to cargo proteins.

The RapiDeg system is based on the rapamycin-induced dimerization between the human FKBP12 protein (FKBP) and the FRB domain of human mTOR (FRB) (*Banaszynski et al., 2005*; *Foster and Fingar, 2010*). In the RapiDeg yeast strain, which is resistant to rapamycin (see 'Materials and methods'), we fused tandem FKBP motifs to Ypq1-GFP, creating Ypq1-GFP-2xFKBP (in short: Ypq1-FKBP). This strain also expresses the FRB domain fused to the N-terminus of a chain of three head-to-tail ubiquitins, which has been proposed to mimic the K63 ubiquitin linkage (*Komander et al., 2009*), creating FRB-3xUb (*Figure 2C*). The addition of rapamycin induces the rapid heterodimerization between FRB-3xUb and Ypq1-FKBP, essentially ubiquitinating Ypq1.

As shown in *Figure 2D and E*, the addition of rapamycin induces rapid sorting of Ypq1-FKBP from the vacuole membrane into the vacuole lumen where it is degraded within 30 min. This is much faster than the degradation kinetics after lysine withdrawal, which takes 4–6 hr, presumably because cells consume their intracellular lysine stores before ubiquitinating Ypq1 (*Li et al., 2015b*). As expected, in the absence of FRB-3xUb, rapamycin treatment does not induce Ypq1-FKBP degradation (*Figure 2—figure supplement 2A,B*). Importantly, the RapiDeg system bypasses the requirement of Ssh4 for Ypq1 ubiquitination (*Figure 2F*).

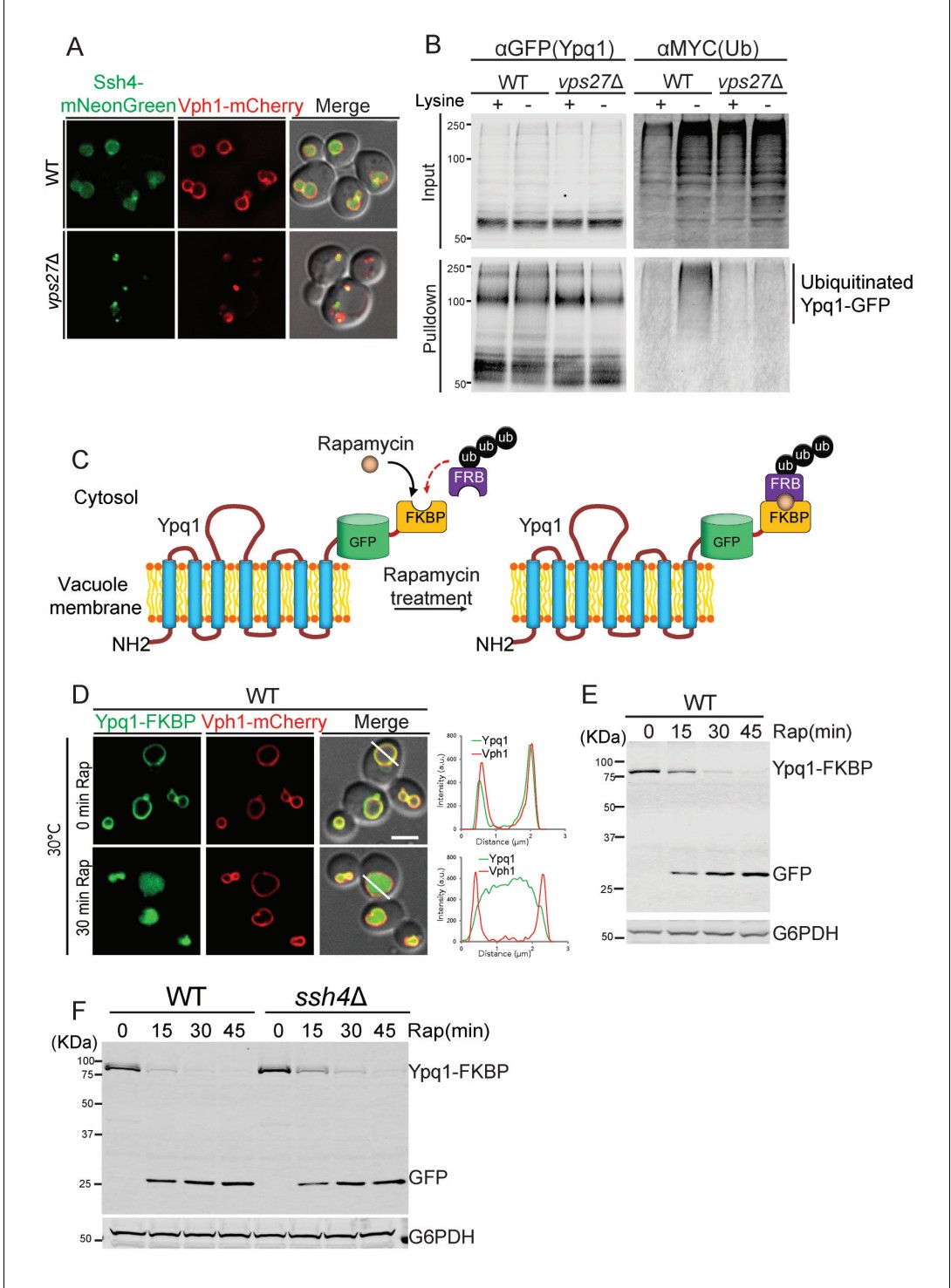

**Figure 2.** Ypq1 ubiquitination defect is bypassed by RapiDeg system. (**A**) Fluorescent microscopy analysis of cells expressing Ssh4-mNeonGreen and Vph1-mCherry in wild-type, and *vps27Δ*(ESCRT-0) cells. (**B**) Ypq1-GFP was immunoprecipitated using GFP-trap resin from *doa4Δ* (WT), and *doa4Δvps27Δ* cells expressing Myc-ubiquitin. Cells were collected before and 2 hr after lysine withdrawal at 26°C. Both input and immunoprecipitated protein samples were analyzed using Western blot and probed with GFP and Myc antibodies. (**C**) A cartoon depicting the RapiDeg system. One (or two) FKBP peptide is fused to the C-terminus of Ypq1-GFP. A chain of three ubiquitins is fused to the C-terminus of the FRB peptide. After adding rapamycin, FRB-3xUb is recruited to Ypq1-FKBP. (**D**) Fluorescent microscopy analysis of the RapiDeg assay. Images show cells co-expressing Ypq1-FKBP and FRB-3xUb before (top) and after rapamycin treatment (1 μg/ml) (bottom) at 30°C. Line scan of both Vph1-mCherry and Ypq1-FKBP was performed using ImageJ (right). (**E**) Western blot analysis of the RapiDeg assay. Whole cell lysates from 0, 15, 30, and 45 min after rapamycin treatment
*Figure 2 continued on next page*

*Figure 2 continued*

(1 µg/ml) at 30°C. Blots were probed with anti-GFP and anti-G6PDH antibodies. (F) Western blot analysis the RapiDeg assay in WT and *ssh4Δ* cells after rapamycin treatment. Blots were probed with anti-GFP and anti-G6PDH antibodies. Scale bars represent 2 µm.

The following figure supplements are available for figure 2:

**Figure supplement 1.** Ssh4 localization and Ypq1 ubiquitination assay in endosome-vacuole fusion mutants.

**Figure supplement 2.** FRB-3xUb is required for rapamycin-induced Ypq1 degradation.

**Figure supplement 3.** The RapiDeg assay is highly specific.

**Figure supplement 4.** The RapiDeg assay can be used to degrade membrane proteins from the PM, Golgi, vacuole, and endosome.

To test the specificity of the RapiDeg system, we added rapamycin to cells co-expressing Ypq1-mCherry and Ypq1-FKBP. As expected, only Ypq1-FKBP was sorted to the vacuole lumen degradation (*Figure 2—figure supplement 3A*). After lysine withdrawal, both Ypq1-FKBP and Ypq1-mCherry were sorted into the vacuole lumen in a Ssh4-dependent manner (*Figure 2—figure supplement 3B,C*). These results suggest that the addition of the FKBP motif does not affect the ability of Ypq1 to respond to lysine withdrawal.

Therefore, by using the RapiDeg system, we can induce the rapid degradation of Ypq1 in the vacuole lumen and thereby study Ypq1 sorting in mutants that would otherwise interfere with the delivery or activity of the Ssh4/Rsp5 ubiquitin ligase complex. Importantly, we also found that the RapiDeg system can be used to study the ubiquitin-dependent trafficking and sorting of transmembrane proteins from the PM (e.g. Can1), Golgi (e.g. Kex2), and endosome (e.g. Nhx1), as well as other vacuole membrane proteins (e.g. Vph1) into the vacuole lumen for degradation (*Figure 2—figure supplement 4*).

## Membrane fusion is not required for Ypq1 sorting

We found that endosome-vacuole fusion mutants block the ubiquitination of Ypq1 after lysine withdrawal. The block of Ypq1 sorting in *vam3ts* or *vps18ts* mutants (*Li et al., 2015b*), therefore, may be due to indirect effects (e.g. Ssh4 sorting block). We set out to examine whether the endosome-vacuole fusion step is required for the sorting of ubiquitinated Ypq1.

To do so, we used the RapiDeg system to test if Ypq1 could be sorted into the vacuole lumen in mutants that block endosome-vacuole fusion. To block endosome-vacuole fusion, we used temperature-sensitive mutants in the CORVET/HOPS complex (*vps18ts allele*) and the SNAP-25-like vacuolar SNARE Vam7 (*vam7ts allele*) (*Rieder and Emr, 1997*; *Sato et al., 1998*). Vps18 is required for the tethering/fusion of biosynthetic, endocytic, and autophagic compartments with the vacuole (*Rieder and Emr, 1997* ; *Robinson et al., 1991*). Vam7, Vam3, and Vti1 co-assemble at the vacuolar membrane into the *t*-SNARE complex, and they are required for the docking and fusion of multiple sorting intermediates destined for the vacuole (*Sato et al., 1998*).

We observed that Ypq1-FKBP is sorted into the vacuole lumen upon cargo ubiquitination using the RapiDeg system in both *vps18ts* and *vam7ts* mutants at the non-permissive temperature (*Figure 3A,C*). These data are consistent with the Western blot results, which demonstrated that Ypq1-FKBP was degraded in the *vps18ts* or *vam7ts* mutants at non-permissive temperatures (*Figure 3B,D*).

To confirm that our temperature shift conditions were sufficient to inactive *vps18ts* and *vam7ts*, we used the endocytic cargo Mup1, which requires endosomal tethering/fusion for delivery and degradation. While the *vps18ts* and *vam7ts* mutants do not block Ypq1 sorting, Mup1 accumulated on pre-vacuolar structures and was not degraded (*Figure 3—figure supplement 1A,B and D*), as expected.

To test more generally whether Ypq1 sorting requires SNARE-mediated membrane fusion, we used a *sec18ts* mutant (*Novick et al., 1980*), which globally inhibits SNARE-mediated trafficking. Sec18 is the yeast NSF (N-ethylmaleimide Sensitive Factor) homolog and is required for disassembly of *cis*-SNARE complexes (*Jun et al., 2007*; *Mayer et al., 1996*). At non-permissive temperature

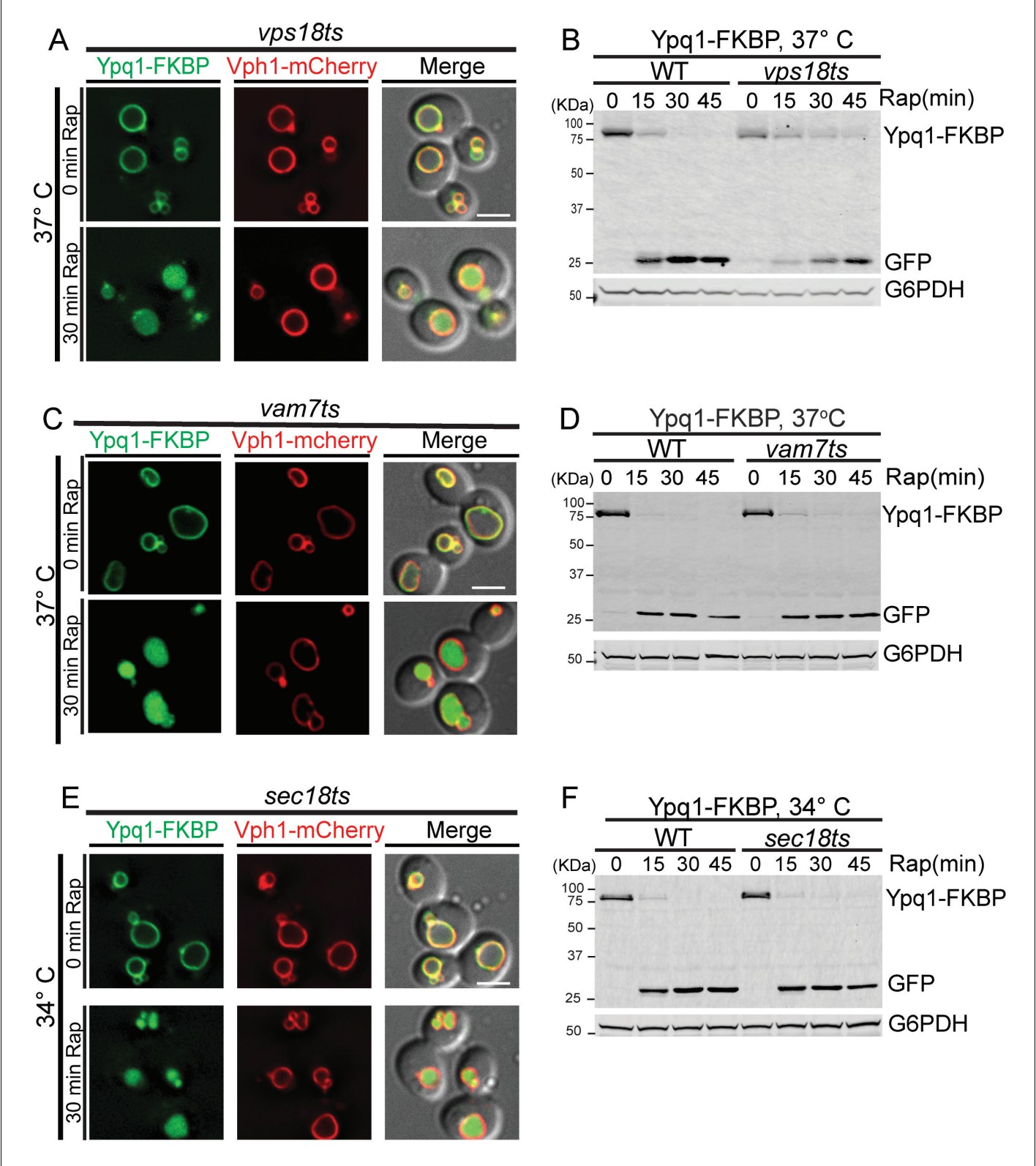

**Figure 3.** Ypq1 sorting does not require membrane fusion. (**A**) Fluorescent microscopy analysis of Ypq1-FKBP and Vph1-mCherry in *vps18ts* cells at non-permissive temperature (37°C). The *vps18ts* cells were grown at 26°C then pretreated at 37°C for 15 min. Images show before (0 min) and after (30 min) rapamycin treatment (RapiDeg). (**B**) Western blot analysis of Ypq1-FKBP degradation. Wild-type and *vps18ts* cells were shifted to 37°C (15 min pretreatment) and collected at four time points (0, 15, 30, and 45 min) after rapamycin treatment. Blots were probed with GFP and G6PDH antibodies. *Figure 3 continued on next page*

*Figure 3 continued*

(C) Fluorescent microscopy analysis of Ypq1-FKBP and Vph1-mCherry in *vam7ts* cells at 37°C. The *vam7ts* cells were grown at 26°C then pretreated at 37°C for 15 min. Images show before and after rapamycin treatment (RapiDeg). (D) Western blot analysis of Ypq1-FKBP degradation. Wild-type and *vam7ts* cells were shifted to 37°C (15 min pretreatment) and collected before and after rapamycin treatment. Blots were probed with GFP and G6PDH antibodies. (E) Fluorescent microscopy analysis of Ypq1-FKBP and Vph1-mCherry in *sec18ts* cells shifted to 34°C (non-permissive temperature). The *sec18ts* cells were grown at 26°C then pretreated at 34°C for 15 min. Images show before and after rapamycin treatment. (F) Western blot analysis of Ypq1-FKBP degradation. Wild-type and *sec18ts* cells were shifted to 34°C(15 min pretreatment) and collected before and after rapamycin treatment. Blots were probed with GFP and G6PDH antibodies. Scale bars represent 2 μm.

The following figure supplement is available for figure 3:

**Figure supplement 1.** Mup1-GFP sorting is blocked in the *sec18ts*, *vps18ts* or *vam7ts* mutants at non-permissive temperature.

(34°C), the *sec18ts* allele does not affect Ypq1 sorting into the vacuole lumen. Western blot analysis confirmed that Ypq1 is degraded in the vacuole lumen with similar kinetics compared to WT cells (*Figure 3E and F*). In contrast, Mup1 endocytosis was blocked in the *sec18ts* mutant (*Figure 3—figure supplement 1A,C*), confirming these conditions inactivate Sec18 function. Thus, these data demonstrate that SNARE-mediated membrane fusion is not required for Ypq1 sorting into the vacuole lumen.

## Ypq1 is sorted into the vacuole lumen by ESCRTs

ESCRT mutants block the ubiquitination of Ypq1 after lysine withdrawal (*Figure 2*), so we used the RapiDeg assay to test whether ESCRTs are required for sorting ubiquitinated Ypq1 on the vacuole membrane.

To test if the sorting of ubiquitinated Ypq1 is affected by ESCRT mutants, we used a *vps4ts* allele to inactivate ESCRT function (*Babst et al., 1997*; *Lata et al., 2008*). After shifting to non-permissive temperature and treating *vps4ts* cells with rapamycin, Ypq1 remained on the vacuole membrane and was not degraded (*Figure 4A,B* and *Figure 4—figure supplement 1A,B*), indicating that the ESCRTs are essential for the sorting of ubiquitinated Ypq1 from the vacuole membrane into the lumen.

Although the ESCRTs have not previously been shown to function on the vacuole membrane, the vacuole contains the lipid PtdIns(3)P and ubiquitinated cargo, both of which are required for ESCRT recruitment to endosomes. The ESCRT-0 subunit Vps27 contains a FYVE domain for PtdIns(3)P binding. To test if the Vps27 FYVE domain binds the vacuole membrane, we generated a GFP-FYVE$_{Vps27}$ reporter and expressed it in WT cells. GFP-FYVE$_{Vps27}$ localized to both vacuolar and endosomal membranes in a Vps34-dependent manner (*Figure 4C*), indicating that the lipid environment of the vacuole is amenable to ESCRT recruitment.

Taken together, these data support a model in which the ESCRT machinery functions at the vacuole membrane during the sorting and degradation of Ypq1. This implies that the ESCRT machinery is recruited to the vacuole membrane following Ypq1 ubiquitination.

## Recruitment of ESCRTs to vacuole membrane for Ypq1 internalization

The data discussed so far suggested a direct role for ESCRTs on the vacuole surface to sort and internalize Ypq1. Such events may be quite fast and difficult to image by light microscopy. Since ESCRT function is ATP dependent (*Babst et al., 1998*; *Tran et al., 2009*), we depleted cellular ATP levels to slow down the internalization process, and thereby help visualize sorting intermediates. We depleted ATP levels using NaN$_3$ and NaF, which block ATP synthesis by oxidative respiration and glycolysis respectively (*Clarke and Weigel, 1985*; *Schmid and Carter, 1990*; *Vida and Emr, 1995*).

We observed that after the recruitment of FRB-3xUb to Ypq1-FKBP in ATP-depleted cells, Ypq1-FKBP sorts into many puncta at the vacuole membrane (*Figure 5A* top and middle). Importantly, Ypq1 does not sort into puncta in the absence of rapamycin (*Figure 5A* bottom), indicating that the puncta are not simply due to the reduction in ATP levels. Moreover, washing out the NaN$_3$ and NaF restored normal Ypq1 sorting (*Figure 5B*). Western blot analysis further confirmed that Ypq1 degradation was blocked by ATP depletion and restored after ATP replenishment (*Figure 5C*). This result

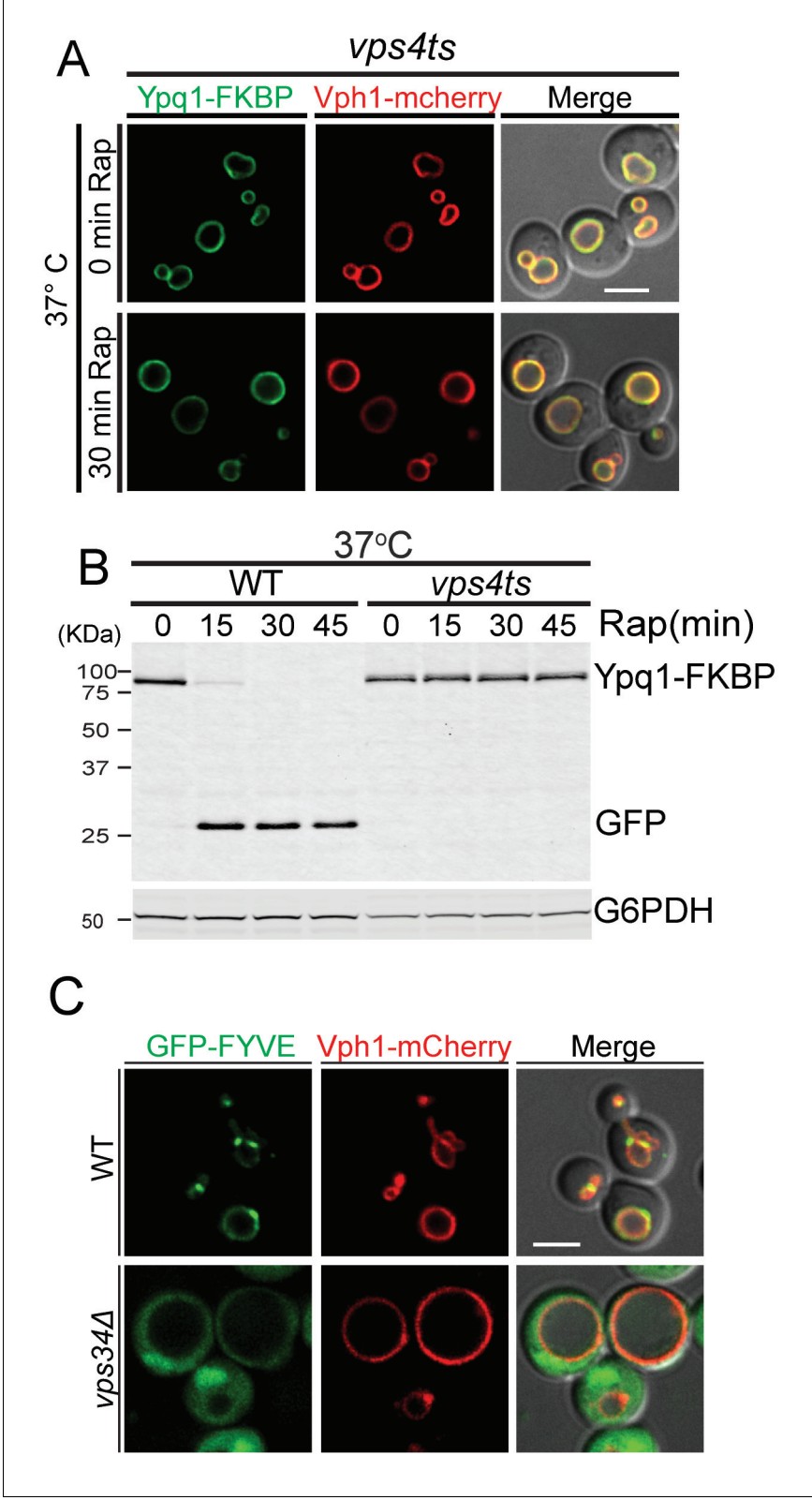

**Figure 4.** In ESCRT mutants, Ypq1 remains on the vacuole membrane after ubiquitination. (**A**) Fluorescent microscopy analysis of Ypq1-FKBP and Vph1-mCherry localization in *vps4ts* cells at 37°C (non-permissive temperature). The *vps4ts* cells were grown at 26°C then pretreated at 37°C for 15 min. Images show before and after rapamycin treatment. (**B**) Western blot analysis of Ypq1-FKBP degradation. Wild-type and *vps4ts* cells were

*Figure 4 continued on next page*

*Figure 4 continued*

shifted to 37°C (15 min pretreatment) and collected before and after rapamycin treatment. The blot was probed with GFP and G6PDH antibodies. (C) Fluorescent microscopy analysis of GFP-FYVE$_{Vps27}$ and Vph1-mCherry in wild-type and *vps34Δ* cells. Scale bars represent 2 μm.

The following figure supplement is available for figure 4:

**Figure supplement 1.** Ypq1-FKBP sorts to the vacuole lumen in the *vps4ts* mutant at permissive temperature.

suggested that the inhibition of ESCRT function by NaN$_3$/NaF treatment is not permanent and can be reversed following ATP recovery.

To test if Ypq1 puncta formation is ESCRT-dependent, we analyzed Ypq1 sorting after ATP depletion in *vps27Δ* cells. After rapamycin and NaN$_3$/NaF treatment, Ypq1-FKBP no longer forms

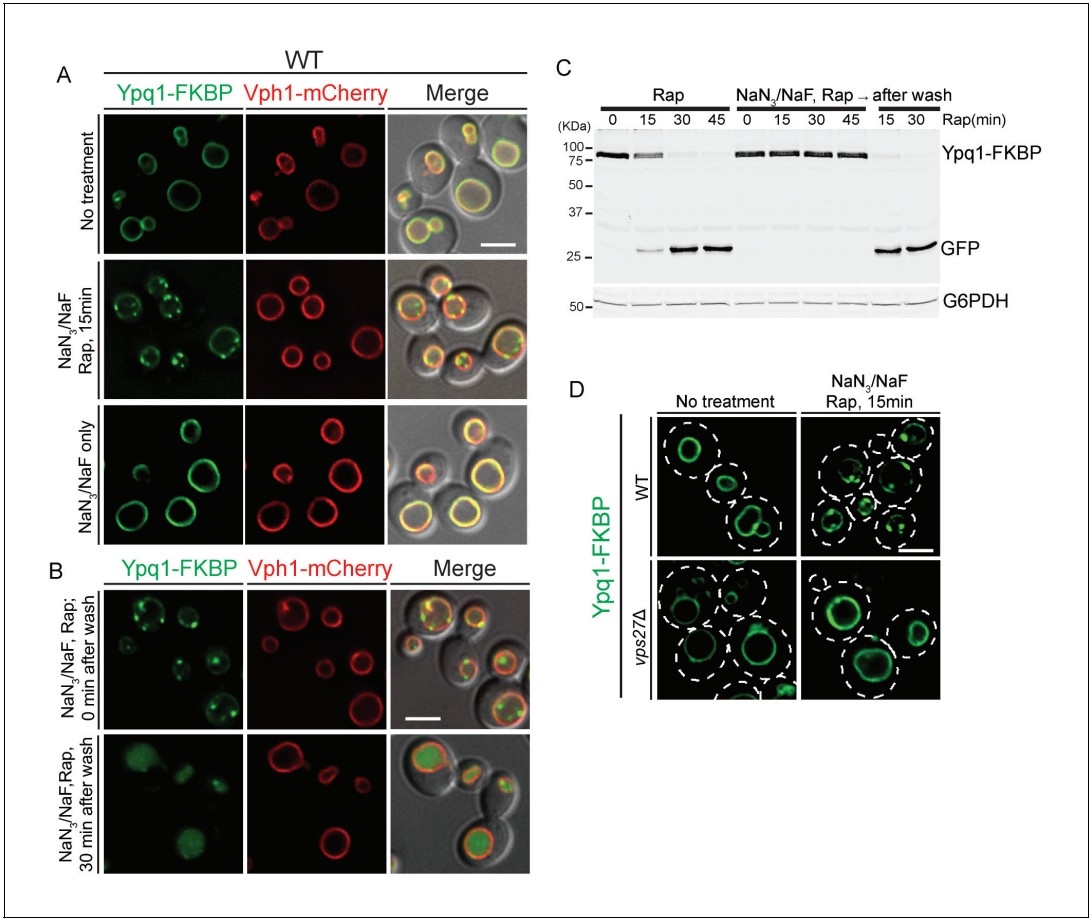

**Figure 5.** ATP-depletion traps ESCRT-dependent Ypq1 sorting intermediates. (A) Fluorescent microscopy analysis of wild-type cells expressing Ypq1-FKBP and Vph1-mCherry. Images show untreated cells (top), Rapamycin (1 μg/ml, pretreated for 1 min) and NaN$_3$/NaF (10 mM each, final concentration) treated cells (middle), and cells treated with only NaN$_3$/NaF. All cells were grown at 30°C (bottom). (B) Fluorescent microscopy analysis of wild-type cells expressing Ypq1-FKBP and Vph1-mCherry. Images shows cells treated with Rapamycin and NaN$_3$/NaF for 30 min (top), and cells from the same culture that have been washed with water and re-suspended in fresh YPD medium (30 min after wash) (bottom). (C) Western blot analysis of Ypq1-FKBP degradation. Wild-type cells were treated with rapamycin, with or without NaN$_3$/NaF. After washing with deionized water and resuspended in fresh YPD medium, the NaN$_3$/NaF-treated cells were collected at 15 min and 30 min. Cells were grown at 30°C. The blot was probed with GFP and G6PDH antibodies. (D) Fluorescent microscopy analysis of Ypq1-FKBP in wild-type and *vps27Δ* cells. Images show cells before and 15 min after rapamycin and NaN$_3$/NaF treatment. Scale bars represent 2 μm.

puncta. Since Vps27 is required for initiating ESCRT-mediated sorting, this suggests that the Ypq1-FKBP sorting intermediates are ESCRT dependent (*Figure 5D*).

To test whether ESCRT proteins are recruited to the vacuole membrane upon Ypq1 ubiquitination, we examined the localization of Ypq1-FKBP and the ESCRT-0 protein Hse1-mCherry (or ESCRT-I protein Vps23-mCherry) after ATP depletion and rapamycin treatment. Before Ypq1 ubiquitination, Ypq1 is localized at the vacuole membrane and Hse1 (or Vps23) is distributed in both the cytosol and on endosomes. Strikingly, after the addition of rapamycin, both Hse1-mCherry and Vps23-mCherry, were recruited to the vacuole and co-localized with the Ypq1-FKBP sorting puncta (*Figure 6A and B*). Similarly, we observed that the ESCRT-II component Vps36-mCherry can also be recruited to vacuole membrane and co-localize with Ypq1-FKBP puncta upon rapamycin treatment and ATP depletion (*Figure 6—figure supplement 1*). Treatment with NaN$_3$/NaF or rapamycin alone, does not lead to detectable co-localization between Ypq1-FKBP and ESCRT proteins (*Figure 6—figure supplement 2*). To address whether the Ypq1-FKBP puncta that form after rapamycin treatment in ATP-depleted cells are endosomes, we stained endosomes with the lipophilic styryl dye FM4-64 (*Vida and Emr, 1995*). We found that the Ypq1-FKBP puncta do not co-localize with FM4-64-positive structures (endosomes) (*Figure 6C*).

Consistent with ATP depletion, we were also able to observe Ypq1 puncta in *vps4ts* cells at non-permissive temperature (*Figure 6—figure supplement 3B*). However, we had to express the ESCRT-0 subunit Hse1 fused with the catalytic domain of the deubiquitinating peptidase UL36 (termed Hse1DUB, *MacDonald et al., 2012*) to release ESCRT proteins from the aberrant endosome (class-E compartment, *Figure 6—figure supplement 3A*), thereby allowing the ESCRT proteins to bind the vacuole and sort Ypq1.

Our observations indicate that the ESCRT machinery can function directly on the vacuole membrane. Essentially, the presence of both ubiquitinated cargo and PtdIns(3)P is sufficient for the recruitment of the ESCRT machinery to both the vacuole membrane and the endosome membrane.

## Visualization of vacuole-membrane-derived intraluminal vesicles

The ESCRT pathway drives membrane remodeling that creates ILVs within the surrounding limiting membrane of the cellular compartment on which ESCRT proteins assemble. Thus, one expectation of vacuolar-localized ESCRT complexes is the formation of ILVs within the vacuole lumen. To detect ILVs that formed from the vacuole membrane, we used electron microscopy to visualize ILVs that formed following rapamycin-induced sorting of Ypq1-FKBP. To avoid contamination of ILVs containing endogenous proteins, we expressed the Hse1DUB to block ILV formation by rapidly deubiquitinating cargoes and thereby preventing ESCRT assembly. The Ypq1-FKBP/FRB-3xUb complex, however, cannot be cleaved by the Hse1DUB, allowing ESCRT assembly and the formation of Ypq1 containing ILVs.

As shown in *Figure 7A*, in the absence of MVB sorting (*pep12Δ*), ILV formation still occurs (Hse1-DUB-/Rap-), although at a much lower level compared to *PEP12* cells (Hse1DUB-/Rap-; *Figure 7D*). The expression of Hse1DUB in *pep12Δ* cells, nearly abolished the generation of ILVs (Hse1DUB+/Rap-; *Figure 7B*). This is in agreement with a previous study which shows that loss of ubiquitinated cargo blocks ILV formation (*MacDonald et al., 2012*). Rapamycin-induced ubiquitination of Ypq1-FKBP caused the accumulation of ILVs in the vacuole lumen (*pep12Δ*, Hse1DUB+/Rap+; *Figure 7C*), as quantified in *Figure 7E*, confirming that Ypq1 is sorted from the vacuole membrane into ILVs by the ESCRT machinery. In addition, the average vesicle size of ILVs formed following rapamycin-induced Ypq1 sorting (~40 nm) were not significantly different from the ILVs formed during ILV formation at the endosome (quantified in *Figure 7F*).

## Discussion

We previously demonstrated that the vacuole membrane transporter Ypq1 is ubiquitinated and then sorted into the vacuole lumen for degradation after lysine withdrawal (*Li et al., 2015a, Li et al., 2015b*). This vacuole membrane protein degradation pathway is critical for the regulation of the quantity and quality of lysosomal membrane proteins in response to intracellular or extracellular cues.

In this study, we applied two genetic approaches to identify the components required for Ypq1 sorting and degradation. We identified an E3 ligase complex (Rsp5/Ssh4), the PtdIns 3-kinase

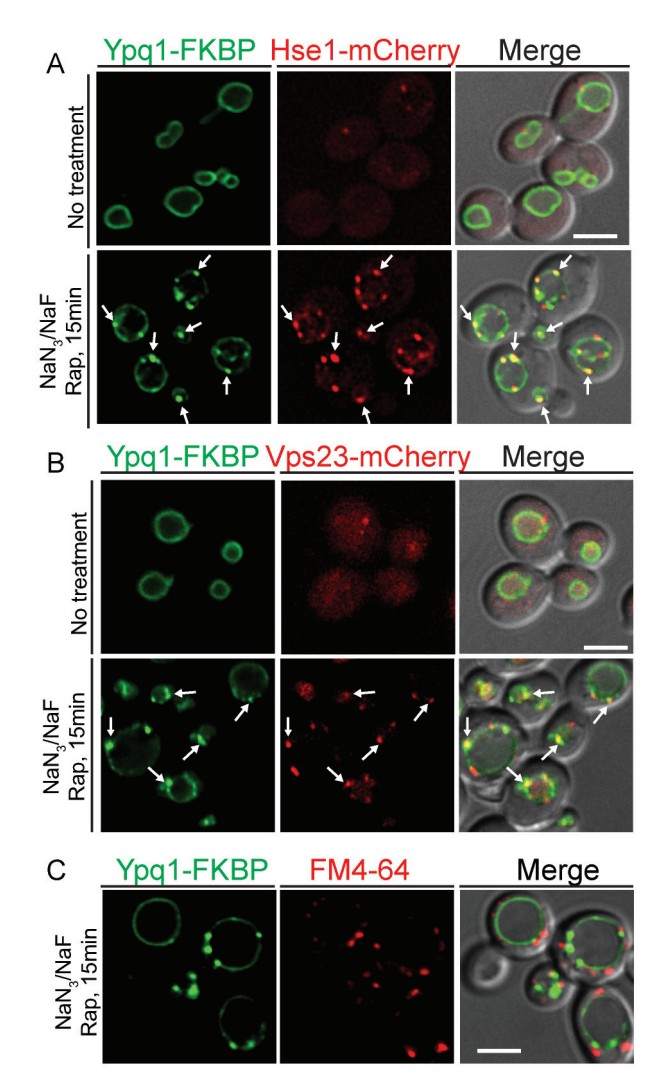

**Figure 6.** ESCRT-0 and -I are recruited to the vacuole membrane after Ypq1 ubiquitination. (**A**) Fluorescent microscopy analysis of Ypq1-FKBP and Hse1-mCherry in wild-type cells. Images show localization of Ypq1-FKBP and Hse1-mCherry before and after rapamycin and $NaN_3$/NaF treatment for 15 min. (**B**) Fluorescent microscopy analysis of Ypq1-FKBP and Vps23-mCherry in wild-type cells. Images show localization of Ypq1-FKBP and Vps23-mCherry before and after rapamycin and $NaN_3$/NaF treatment. (**C**) Fluorescent microscopy analysis of Ypq1-FKBP and FM4-64 (endosome dye) in wild-type cells after rapamycin and $NaN_3$/NaF treatment at 30°C. Cells were stained with FM4-64 for 5 min prior to rapamycin and $NaN_3$/NaF treatment. Arrows indicate representative co-localization between Ypq1-FKBP and Hse1-mCherry (or Vps23-mCherry). Scale bars represent 2 μm.

The following figure supplements are available for figure 6:

**Figure supplement 1.** ESCRT-II protein Vps36 is recruited to the vacuole membrane after Ypq1 ubiquitination.

**Figure supplement 2.** Ypq1-FKBP and Vps23-mCherry do not co-localize after $NaN_3$/NaF or rapamycin treatment alone.

**Figure supplement 3.** Expression of Hse1DUB, releases ESCRTs from endosomes, allowing for ESCRTs recruitment onto the vacuole membrane in the *vps4ts* mutant.

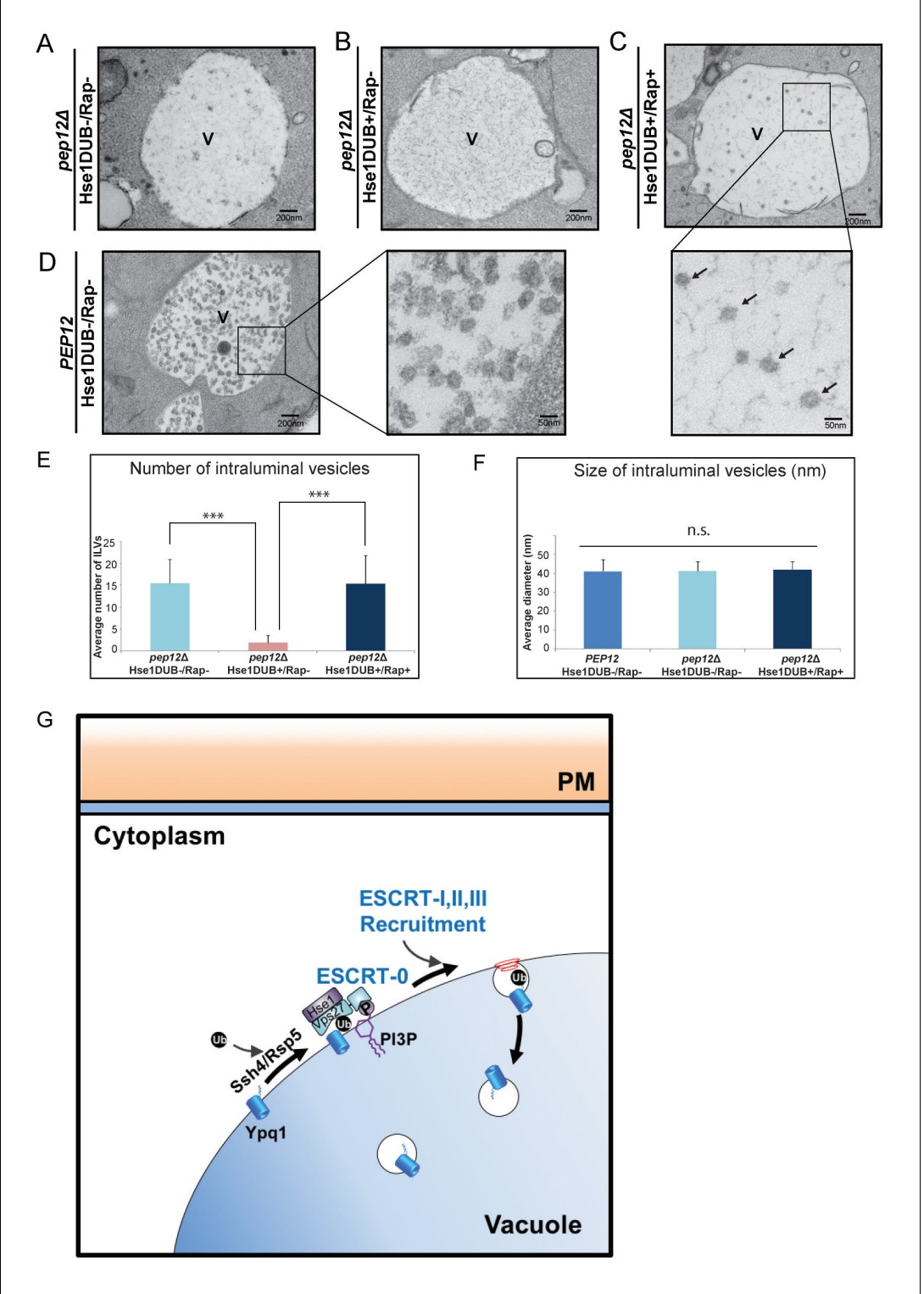

**Figure 7.** Visualization of Ypq1-FKBP ILVs. (**A–B**) EM images of the vacuole in *pep12Δpep4Δvma4Δatg9Δ* cells (*pep12Δ*) with or without Hse1DUB (No Rapamycin treatment). (**C**) EM image of the vacuole in cells expressing Hse1DUB after rapamycin treatment (2 hr) at 26°C. Arrows indicate representative vesicles formed following rapamycin-induced Ypq1 sorting. (**D**) EM images of the vacuole in *pep4Δvma4Δatg9Δ* cells (*PEP12*) without Hse1DUB (No Rapamycin treatment). 'V' in the figures (7A~D) is abbreviated for vacuole. (**E**) Average number of ILVs (per cell per section) calculated by counting ILVs within vacuoles of cells (*pep12Δ*) without Hse1DUB, with Hse1DUB, and with both Hse1DUB and rapamycin treatment. The mean value of ILVs formed in vacuole lumen

*Figure 7 continued on next page*

*Figure 7 continued*

was quantified and error bars are standard deviation. Two-tail t-test was performed between *pep12Δ*(Hse1DUB-/Rap-) and *pep12Δ*(Hse1DUB+/Rap-), and between *pep12Δ*(Hse1DUB+/Rap-) and *pep12Δ*(Hse1DUB+/Rap+). $\alpha$ = 0.025 (Bonferroni correction), p-value<0.001 in each test (***, highly significant), N = 40 cells for each condition. (F) Average diameter of ILVs in cells (*PEP12*) without Hse1DUB or rapamycin treatment, and in cells (*pep12Δ*) without Hse1DUB or rapamycin treatment, as well as cells (*pep12Δ*) with both Hse1DUB and rapamycin treatment. The mean value of ILV size formed in the vacuole lumen was quantified and error bars are standard deviation. ANOVA (single factor) test was performed among *PEP12*(Hse1DUB-/Rap-), *pep12Δ*(Hse1DUB-/Rap-), and *pep12Δ*(Hse1DUB+/Rap+). $\alpha$ = 0.025 (Bonferroni correction), p-value>0.05 (n.s., non-significant), N = 64 vesicles for each condition. (G) Cartoon model depicting Ypq1 sorting from the vacuole membrane into the vacuole lumen. After lysine withdrawal, Ypq1 is ubiquitinated by the Ssh4/Rsp5 E3-ligase complex. Ubiquitinated Ypq1 and PtdIns(3)P recruit ESCRT-0 to the vacuole membrane. ESCRT-0 sorts Ypq1 and recruits downstream ESCRTs to the vacuole membrane. The ESCRTs internalize Ypq1 directly into the vacuole lumen for degradation.

complex (Vps15/Vps34), the ESCRT complexes, and the endosome-vacuole fusion machinery (HOPS/CORVET/SNAREs). After further characterization, we found that Ypq1 cargo ubiquitination is blocked in ESCRT and endosome-vacuole fusion mutants due to missorting of Ssh4, the ubiquitin ligase adaptor protein normally found on the vacuole. We therefore developed the RapiDeg system to bypass the Ssh4 requirement for Ypq1 ubiquitination by conjugating ubiquitin to Ypq1 in a rapamycin-dependent manner. Using this system, we have shown that fusion between the vacuole and an intermediate compartment (endosome) is not required for Ypq1 delivery to the vacuole lumen. The ESCRT complexes, however, are absolutely required for Ypq1 sorting. Moreover, the ESCRTs can be recruited to the vacuole membrane to mediate the sorting of Ypq1 upon Ypq1 ubiquitination. We therefore propose that ubiquitinated vacuole membrane proteins are sorted directly on the vacuole membrane by the ESCRTs, which then internalize the cargo into ILVs that bud into the vacuole lumen.

This model is in contrast with our previously proposed model (*Li et al., 2015b*) in which we proposed that Ypq1 is sorted from the vacuole membrane to an intermediate compartment (endosome) before being delivered to the vacuole lumen. The previous model was based on observations that Ypq1-GFP formed puncta in ~13% of cells after 2 hr lysine starvation, and a small number of the Ypq1-GFP puncta (9 Ypq1-GFP puncta per ~500 cells were detected) were detached from the vacuole membrane after lysine withdrawal. Based on our current study, we now believe that all the puncta, which were attached to the vacuole membrane, represent sorting intermediates on the vacuole membrane destined for direct internalization into the vacuole lumen.

While we cannot rule out the possibility that some Ypq1 is first sorted from the vacuole membrane to an intermediate compartment, the development of the RapiDeg system has allowed us to better test the requirements of this sorting pathway. We have shown that Ypq1 sorting does not require an intermediate compartment, and it appears that the vast majority of protein is sorted directly from the vacuole membrane into the vacuole lumen.

Recently, a new model for selective sorting and degradation of vacuole membrane proteins has been proposed, the intralumenal fragment (ILF) pathway (*McNally et al., 2017*). In this model, vacuolar membrane proteins are sorted into the interface of two opposing vacuoles and are internalized after vacuole fusion in an ESCRT-independent manner. We do not think that this pathway plays a significant role in ubiquitin-dependent sorting of Ypq1, because Ypq1 sorting is ESCRT dependent, and does not require Sec18(NSF), Vps18(HOPS) and Vam7(Vacuole SNARE), which are required for vacuole-vacuole fusion (*Wickner, 2010*).

## RapiDeg system

While we developed the RapiDeg system to study ubiquitin-dependent sorting of Ypq1, we have also shown that this system provides a simple and efficient method to induce ubiquitin conjugation to other membrane proteins. RapiDeg can be used to induce the trafficking and degradation (inactivation) of membrane proteins in the secretory (Golgi) and endocytic (PM and endosome) pathways. The two ER membrane proteins we tested were not degraded using the RapiDeg system (Erd2, Heh2).

What Lys-linkage does the head-to-tail FRB-3xUb fusion used in the RapiDeg system resemble—K48 or K63? It was previously reported that K63-linked and linear head-to-tail ubiquitins adopt equivalent open conformations, in which the ubiquitins do not form inter-molecular interactions like K48-linked ubiquitin chains (*Komander et al., 2009*). It was also found that ESCRT-0 complexes preferably bind to K63-linked ubiquitin chains (*Nathan et al., 2013*). This may explain why the conjugates of head-to-tail FRB-3xUb fused to PM, endosome, vacuole, or Golgi membrane proteins results in their sorting, via the ESCRT machinery, into the vacuole lumen.

Therefore, the RapiDeg system provides a tractable method for the specific inactivation (degradation) of membrane proteins in the secretory and endocytic pathways. It will also be useful for studying ubiquitin dependent sorting of membrane proteins into the vacuole lumen. Importantly, RapiDeg should be adaptable for use in a wide range of eukaryotic organisms.

## Recruitment of ESCRTs to the vacuole membrane

ESCRT recruitment to the vacuole membrane uses the same components as ESCRT recruitment to the late endosome membrane, PtdIns(3)P and ubiquitinated cargo. While these environments may be similar to ESCRTs, the exact composition of vacuole membrane derived ILVs is of particular interest. The vacuole membrane is inherently resistant to its luminal lipases and proteases, while ILVs derived from this membrane, must be lysed in order to degrade their protein cargoes. We are currently investigating the specific vulnerabilities of vacuole membrane-derived ILVs, specifically: whether certain lipids are included or excluded from ILVs; whether protein concentration de-stabilizes ILVs; and whether the high curvature of ILVs allows vacuolar lipases to lyse them.

While we have shown that the ESCRTs function on the vacuole membrane, we could not detect a vacuolar ESCRT pool using fluorescent microscopy under normal conditions. We think the reason for this is twofold. First, the endosome is the major site for ILV formation due to the high concentration of ubiquitinated cargo arriving from the PM and the Golgi. Second, the large surface area of the vacuole reduces local ESCRT concentrations below our present detection limits.

How could ATP depletion allow us to visualize Ypq1 sorting puncta, and ESCRT recruitment to the vacuole membrane after rapamycin treatment? Primarily, we reason that these puncta represent ESCRT sorting complexes which have stalled at an intermediate step due to loss of function of Vps4, an ATP-dependent and essential ESCRT component. Further, these sorting intermediates may be stabilized because ubiquitination of RapiDeg cargo cannot be removed by deubiquintinases, causing early ESCRTs to remain engaged with cargo. Moreover, it is possible that stalled sorting complexes can cluster together, this may be due to the ability of the ESCRT-0 complex to form oligomers (*Mayers et al., 2011*; *Wollert and Hurley, 2010*).

## Multiple sites and roles for the ESCRT machinery

The ESCRTs were first characterized as cargo sorting machinery required for ILV formation at the endosome (*Henne et al., 2011*; *Odorizzi et al., 1998*). Recently, the ESCRTs have been shown to function on various other cellular membranes. These roles include HIV budding at the PM (*Garrus et al., 2001*; *Martin-Serrano et al., 2001*; *VerPlank et al., 2001*), PM wound repair (*Jimenez et al., 2014*), nuclear-pore assembly (*Webster et al., 2014*), nuclear envelope re-formation (*Olmos et al., 2015*; *Vietri et al., 2015*; *Webster et al., 2014*), and cytokinesis (*Carlton and Martin-Serrano, 2007*; *Lee et al., 2008*). Here, the turnover of vacuole membrane proteins revealed yet another subcellular location where ESCRTs function.

In summary, we propose the following model (*Figure 7G*). The combination of PtdIns(3)P and ubiquitinated Ypq1 on the vacuole membrane specifies the recruitment of the ESCRT-0 complex to the vacuole membrane. This event initiates the recruitment of downstream ESCRT complexes to the vacuole membrane, and ultimately results in the sorting of these cargoes into intraluminal vesicles that bud directly into the vacuole lumen.

## Materials and methods

### Yeast strains, plasmids, and cell growth conditions

All the strains and plasmids used in this study are documented in the *Supplemental file 3A and 3B*. For Ypq1 degradation induced by lysine starvation, cells were grown overnight to mid log phase

(OD600 ~ 0.5) in synthetic media containing 23 µg/ml lysine at 26℃ then incubated in lysine-free media for 6 hr (*Li et al., 2015b*). For RapiDeg, cells were grown in YPD media or SD-media to mid-log phase (OD600 ~0.5) at 26℃ before adding rapamycin (final concentration: 1 µg/ml). After rapamycin treatment, cells were collected at different time points for imaging or whole cell lysate extraction. For temperature-sensitive strains, cells were grown at 26℃, then shifted to 34℃ (for *sec18ts*) or 37℃ (for *vps18ts*, *vam7ts* and *vps4ts*) for 15 min before rapamycin treatment. The mNeonGreen gene used for the Ssh4 visualization is licensed by Allele Biotechnology (San Diego) (*Shaner et al., 2013*).

## Spontaneous mutagenesis and whole-genome visual screen for Ypq1 missorting mutants

For the spontaneous mutagenesis selection, Ypq1-SBP-GFP-His3 was integrated into the *LEU2* locus of SEY6210 *MATα* and SEY6211 *MATa*. Single colonies were patched onto -His media containing 5 mM 3-Amino-1,2,4- triazole (3-AT, Sigma-Aldrich, St. Louis, MO) and grown for 10 days to generate spontaneous His+ mutants. 150 *MATα* and 150 *MATa* His+ mutants were isolated and mated into 18 complementation groups. Each group was then identified using plasmid complementation (URA marker, 2µ, [*Herman et al., 1991*]) or by mating to tester strains.

For the genome wide visual screen, pML359 was transformed into the BY4741 non-essential gene knock-out collection using a large scale 96-well transformation method (*Gietz and Schiestl, 2007*). The transformants were selected on SD-URA plates and grown in SD-URA media to log-phase. The fluorescence of each mutant expressing Ypq1-SBP-pHluorin was quantified using flow-cytometry. We used *ssh4Δ* mutant as positive controls and WT cells as negative controls. BY4741 cells not expressing fluorescent proteins was used per each scan as a background control to normalize each mutant fluorescent intensity. The dataset for 'fold change of fluorescence intensity for each mutant (compared with WT cells)' was attached in the *Supplemental file 1*. Ypq1-localization was further validated for sorting mutants using fluorescence-microscopy.

## Construction of the RapiDeg yeast strain

WT strains (SEY6210 or SEY6210.1) were made rapamycin-resistant by transforming a PCR fragment encoding the dominant *TOR1* mutant (S1972I) and selected for growth on YPD plates (supplemented with 1 µg/ml rapamycin). *FPR1* (*FKBP-12*) was knocked out using homologous recombination to ensure pairing of FRB-3xUb to target proteins tagged with FKBP. To prevent the conjugation of FRB-3xUb to endogenous proteins and cleavage by deubiquitinating enzymes, Gly75 and Gly76 of the first and second ubiquitins were replaced with Ala-Ala-Arg-Ser and Gly76 of the third ubiquitin was replaced with Glu.

## Whole cell lysate extraction and western blotting

Whole cell extracts were prepared by incubating 5 ODs of cells in 10% Trichloroacetic acid on ice for 1 hr. Extracts were fully resuspended with ice-cold acetone twice by sonication, then vacuum-dried. Dry pellets were mechanically lysed (4 × 5 min) with 100 µL glass beads and 100 µL Urea-Cracking buffer (50 mM Tris.HCl pH 7.5, 8 M urea, 2% SDS, 1 mM EDTA). 100 µl protein sample buffer (150 mM Tris.HCl pH 6.8, 7 M urea, 10% SDS, 24% glycerol, bromophenol blue) supplemented with 10% 2-mercaptoethanol was added and samples were vortexed for 5 min. The protein samples were resolved on 11% SDS-polyacrylamide gels and then transferred to nitrocellulose blotting membranes (GE Healthcare Life Sciences). Antibodies to G6PDH (Sigma-Aldrich, rabbit, 1:30,000) and GFP (Santa Cruz, mouse, 1:1,000) were used for the primary binding. After incubation with secondary antibodies (IRDye 800CW Goat anti-Mouse IgG and IRDye 680RD Goat anti-Rabbit IgG, 1:10,000 for each), the proteins were visualized using Li-Cor system. Westernblot experiments were performed at least three times from at least three biological replicates.

## Ubiquitination assay

The ubiquitination assay was adapted from *Li et al. (2015b)*. In brief, cells were grown to early log phase in synthetic media. Myc-Ub expression was induced 4 hr prior to lysine withdrawal by treating cells with 100 µM $CuSO_4$. 60 ODs of cells were harvested after 2 hr lysine starvation, washed once with water, then snap frozen in liquid nitrogen. The pellets were resuspended with 500 µL IP buffer

(50 mM HEPES-KOH, pH 6.8, 150 mM KOAc, 2 mM MgOAc, 1 mM CaCl2, and 15% glycerol) with cOmplete protease inhibitor (Sigma-Aldrich, St. Louis, MO) and 20 mM N-ethylmaleimide. Cell extracts were prepared by adding 400 µL glass beads and beating for 10 min at 4°C. Membrane proteins were solubilized by adding 500 µL of 2% digitonin in IP buffer. Cell lysates were clarified by spinning at 34,000xg for 15 min at 4°C. The resulting lysate was then incubated with 20 µL GFP-nanotrap resin for 2 hr at 4°C. The resin was washed 4 times with 0.5% digitonin in IP buffer. Bound protein was eluted by adding sample buffer and incubating for 10 min at 65°C.

## Fluorescent and electron microscopy

For fluorescence microscopy, cells expressing GFP fusion or mCherry proteins were visualized using a DeltaVision Elite system (GE), equipped with a Photometrics CoolSnap HQ2/sCMOS Camera, a 100×objective, and a DeltaVision Elite Standard Filter Set ('FITC' for GFP/pHluorin fusion protein and 'mCherry' for mCherry fusion proteins). Image acquisition and deconvolution were performed using Softworx. GFP/mCherry images were merged using NIH ImageJ-win64. Microscopy experiments were performed at least three times from at least three biological replicates.

Electron microscopy on chemically fixed yeast cells was performed as previously described (*Buchkovich et al., 2013*) with minor modification. Yeast cells were fixed with glutaraldehyde (3%, v/v) for 1 hr and spheroplasted with zymolyase (200 µg/ml) and beta-glucuronidase (10%) then embedded in the 2% (m/v) ultra-low gelling temperature agarose. Cells were stained with 1% ferrocyanide-reduced osmium (30 min), followed by 1% thiocarbohydrazide (5 min) before re-stained with 1% ferrocyanide-reduced osmium (5 min). Cells were washed with Milli-Q water four times between each treatment. After ethanol dehydration, the cells samples were placed in propylene oxide and finally embedded in Epon resin. All electron microscopy was imaged with AMT digital camera equipped on a Morgnani 268 transmission electron microscope (FEI).

## Acknowledgements

We thank Drs. Chris J Fromme, Chun Han and William J Brown for helpful advice. We also thank Anthony Gatts for technical assistance of Electron Microscopy. We are grateful to Drs. William M Henne and Matthew G Baile for critical reading of the manuscript. We also thank Drs. Sho Suzuki, Richa Sardana, Hsuan-chung Ho, Evan L Guiney, Till Klecker and other members of the Emr lab for helpful discussions. We appreciate the spiritual encouragement and scientific guidance by Dr. Jeremy W Thorner. Ya-shan Chuang is supported by a scholarship from the Ministry of Education (Taiwan). The work was supported by a Cornell University Research Grant (SDE).

## Additional information

### Funding

| Funder | Grant reference number | Author |
| --- | --- | --- |
| Cornell University | Research Grant CU3704 | Scott D Emr |

The funders had no role in study design, data collection and interpretation, or the decision to submit the work for publication.

### Author contributions

LZ, JRJ, Conceptualization, Data curation, Formal analysis, Validation, Investigation, Visualization, Methodology, Writing—original draft, Writing—review and editing; ML, Conceptualization, Methodology, Writing—review and editing, Dr. Ming Li and Mr. Ya-shan Chuang developed the RapiDeg assay in this study; Y-SC, Methodology, Dr. Ming Li and Mr. Ya-shan Chuang developed the RapiDeg assay in this study; SDE, Conceptualization, Resources, Supervision, Funding acquisition, Project administration, Writing—review and editing

## Author ORCIDs
Lu Zhu, http://orcid.org/0000-0001-9438-9296
Jeff R Jorgensen, http://orcid.org/0000-0003-1009-9959
Scott D Emr, http://orcid.org/0000-0002-5408-6781

## Additional files

### Supplementary files
• Supplementary file 1. Compiled fluorescence fold change data form each mutant in the fluorescence based screen.

• Supplementary file 2. The mutants defective for Ypq1 sorting found by 'spontaneous mutagenesis' and 'fluorescence screen'

• Supplementary file 3. Supplementary experimental materials.

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
