## [Decision Letter]

Thank you for submitting your article "ESCRTs function directly on the lysosome membrane to downregulate ubiquitinated lysosomal membrane proteins" for consideration by *eLife*. Your article has been reviewed by 3 peer reviewers and the evaluation has been overseen by Randy Schekman as the Reviewing and Senior Editor. The following individuals involved in review of your submission have agreed to reveal their identity: Randy Schekman (Reviewer #1) and Alexey J. Merz (Reviewer #3).

The reviewers have discussed the reviews with one another and the Reviewing Editor has drafted this decision to help you prepare a revised submission.

Zhu et al., present compelling evidence that vacuolar transporters, particularly the Lys transporter Ypq1, are turned over by internalization directly into the vacuole mediated by the ESCRT machinery. To make this claim, they devised growth and fluorescent reporters to screen the non-essential yeast gene library to identify candidates for this turnover pathway. Among the genes turned up, ESCRT, PI3kinase and vacuole-endosome fusion proteins were identified. Importantly, they also devised a clever test of the direct or indirect role of these genes by creating a rapamycin-induced ubiquitylation tag that bypasses the requirement for upstream functions that may be necessary to localize Ssh4, the Rsp5 ubiqitin ligase receptor, to the vacuole membrane. Using this RapiDeg approach, the authors show that the vacuole-endosome fusion functions act upstream of the ESCRT machinery, which they argue acts directly on ubiquitylated Ypq1 to promote internalization into the vacuole. These results are extended to demonstrate that ESCRT proteins can be recruited directly to the vacuole surface dependent on the action of the PI3 kinase and that ESCRT punctae can be observed to form on the vacuole surface in situations where the internalization process is delayed in cells incubated with energy poisons.

Essential revisions:

1) The Emr group published a different conclusion two years ago with the claim that Ypq1 is withdrawn from the vacuole surface to an endosome-like organelle into which ubiquitylated Ypq1 is internalized followed by fusion of the MVB back to the vacuole (Li et al., 2015).

To be fair, the previous work established many of the basic facts upon which the current work has relied for more incisive analysis. However at least some of the work in that previous paper would appear to be contradicted by results in the current manuscript. For example:

"To directly test if the recycling hypothesis is correct, we used a strain (YYR36, Table S1) co-expressing Ypq1-GFP and Vph1-mCherry from the chromosome to investigate if lysine withdrawal can trigger a selective sorting of Ypq1-GFP. Indeed, after lysine withdrawal, we observed selective sorting of Ypq1-GFP into a compartment that did not co-localize with Vph1-mCherry (Figure 6). Many of these vacuole membrane-attached compartments appeared to be tethered by the HOPS complex, as indicated by the colocalization of Ypq1-GFP with Vps18-mCherry and their movement together on the vacuole surface (Figure 5 and Movie S1). This was consistent with the essential role of Vps18 for Ypq1 degradation and further suggested that tethering the Ypq1-GFP compartment to the vacuole is a critical intermediate step."

How can this previous observation be reconciled with the current results of a direct internalization into the vacuole? I appreciate that the lysine withdrawal approach used previously results in a much slower degradation of Ypq1 than the RapiDeg trick developed here affords, and that may be explained by a more indirect effect that progresses over the prolonged incubation necessary to observe the lysine withdrawal effect. However, one could argue that lysine deprivation is the more natural path for Ypq1 degradation.

Nonetheless, the current results are highly original and persuasive and certainly merit publication in *eLife*, but the authors must do a better job in placing the new results in the context of their previous publication. As it stands, the authors offered no explanation of the discrepancy with their previous paper. They owe the readers who are knowledgeable in this area a more thorough treatment of the subject. This issue should be highlighted in the Introduction and then explained fully in the Discussion section of a revised manuscript.

2) The papers from Brett's lab (http://dx.doi.org/10.1091/mbc.E15-11-0759 and http://dx.doi.org/10.1016/j.devcel.2016.11.024) need to be cited and discussed. Particularly the second of these papers describes a completely different, ESCRT-independent, pathway for the turnover of vacuolar proteins. Did the Brett group use different vacuolar membrane proteins in their investigation? If not, then how might the two results be reconciled?

3) Subsection “ESCRT and endosome-vacuole fusion mutants block Ypq1 ubiquitination”, "Ssh4 delivery appeared to require this fusion machinery". This statement lacks data support. Furthermore, the *ts* mutants of the fusion machinery are supposed to not affect Ssh4 molecules that have already reached vacuole before temperature shift. Are there explanations for why preexisting Ssh4 on the vacuole membrane is not sufficient to support Ypq1 ubiquitination after temperature shift?

4) The strongest line of evidence against the direct invagination hypothesis in Li et al., 2015 is the fact that upon lysine withdraw, Ypq1 was observed on structures detached from the vacuole, including: Ypq1-GFP punctae detached from the vacuole surface (Figure 6 and Figure S5B in Li et al.,), Ypq1-nGFP and Vps23-cGFP split-GFP punctae detached from the vacuole surface (Figure S4 in Li et al.,) and Ypq1-nGFP and cGFP-Ub split-GFP punctae detached from the vacuole surface (Figure S6D in Li et al.). The authors should discuss possible explanations of these earlier observations. Is it possible that Ypq1-SBP internalization and RapiDeg-induced Ypq1 internalization mainly go through the direct invagination route, whereas lysine withdrawal-induced Ypq1 internalization at least partially goes through an intermediate compartment?

5) The authors show persuasively that brief shift to nonpermissive temperature for *vps18* and *vam3 tsf* mutants prevents Ypq1 ubiquitination upon lysine withdrawal. They argue (Results section) that this is due to the absence of Ssh4 on the vacuole, but that key inference is not tested by looking at Ssh4 localization. It should be.

6) As a control for the blocks imposed by docking and fusion *tsf* mutants, Mup1 traffic through the MVB to the vacuole is examined. This is a good control but it involves a heterotypic rather than homotypic vacuole fusion event. Adding to this concern, it's a bit surprising to see wt/class A non-fragmented vacuole morphologies in the *vps18tsf* and *vam7 tsf* mutants after 45' (Figure 3) or 75' (Figure 3—figure supplement 1) at nonpermissive temperature.

7) Since rapamycin treatment shuts off TORC1 with a cascade of effects on protein synthesis degradation, and stress signaling, more controls are critical. Missing controls include: monitoring of substrate reporters with mutant rapamycin non-binding FKBP domain; absence of FRB-Ub3; and/or the use of rapamycin-insensitive TOR1 alleles.

---

## [Author Response]

*1) The Emr group published a different conclusion two years ago with the claim that Ypq1 is withdrawn from the vacuole surface to an endosome-like organelle into which ubiquitylated Ypq1 is internalized followed by fusion of the MVB back to the vacuole (Li et al., 2015).*

*To be fair, the previous work established many of the basic facts upon which the current work has relied for more incisive analysis. However at least some of the work in that previous paper would appear to be contradicted by results in the current manuscript. For example:*

*"To directly test if the recycling hypothesis is correct, we used a strain (YYR36, Table S1) co-expressing Ypq1-GFP and Vph1-mCherry from the chromosome to investigate if lysine withdrawal can trigger a selective sorting of Ypq1-GFP. Indeed, after lysine withdrawal, we observed selective sorting of Ypq1-GFP into a compartment that did not co-localize with Vph1-mCherry (Figure 6). Many of these vacuole membrane-attached compartments appeared to be tethered by the HOPS complex, as indicated by the colocalization of Ypq1-GFP with Vps18-mCherry and their movement together on the vacuole surface (Figure 5 and Movie S1). This was consistent with the essential role of Vps18 for Ypq1 degradation and further suggested that tethering the Ypq1-GFP compartment to the vacuole is a critical intermediate step."*

*How can this previous observation be reconciled with the current results of a direct internalization into the vacuole? I appreciate that the lysine withdrawal approach used previously results in a much slower degradation of Ypq1 than the RapiDeg trick developed here affords, and that may be explained by a more indirect effect that progresses over the prolonged incubation necessary to observe the lysine withdrawal effect. However, one could argue that lysine deprivation is the more natural path for Ypq1 degradation.*

*Nonetheless, the current results are highly original and persuasive and certainly merit publication in eLife, but the authors must do a better job in placing the new results in the context of their previous publication. As it stands, the authors offered no explanation of the discrepancy with their previous paper. They owe the readers who are knowledgeable in this area a more thorough treatment of the subject. This issue should be highlighted in the Introduction and then explained fully in the Discussion section of a revised manuscript.*

We highlight the data that led to the recycling model in the Introduction and explain these data in the Discussion.

*2) The papers from Brett's lab (http://dx.doi.org/10.1091/mbc.E15-11-0759 and http://dx.doi.org/10.1016/j.devcel.2016.11.024) need to be cited and discussed. Particularly the second of these papers describes a completely different, ESCRT-independent, pathway for the turnover of vacuolar proteins. Did the Brett group use different vacuolar membrane proteins in their investigation? If not, then how might the two results be reconciled?*

Brett’s group has characterized a pathway in which vacuole membrane proteins are sorted into the interface of two opposing vacuoles, and are internalized after vacuole fusion in an ESCRT independent manor. They did not use Ypq1 when characterizing this pathway. Further, since Ypq1 degradation requires ESCRTs, and does not require the vacuole-vacuole fusion machinery, it appears that ubiquitin dependent turnover of vacuole membrane proteins uses a distinct pathway from the “Intra-Luminal Fragment pathway”. We have discussed the ILF pathway and its relation to Ypq1 sorting in the Discussion.

*3) Subsection “ESCRT and endosome-vacuole fusion mutants block Ypq1 ubiquitination”, "Ssh4 delivery appeared to require this fusion machinery". This statement lacks data support. Furthermore, the ts mutants of the fusion machinery are supposed to not affect Ssh4 molecules that have already reached vacuole before temperature shift. Are there explanations for why preexisting Ssh4 on the vacuole membrane is not sufficient to support Ypq1 ubiquitination after temperature shift?*

We have included imaging data that shows that Ssh4 accumulates in pre-vacuolar compartments in both *vam3ts* and *vps18ts* cells at non-permissive temperature (Figure 2—figure supplement 1). Our data suggest that biosynthetic Ssh4 is required, and that the pre-existing pool of Ssh4 may be insufficient for Ypq1 ubiquitination (subsection “ESCRT and endosome-vacuole fusion mutants block Ypq1 ubiquitination“). In addition, since Ssh4 localizes to both the vacuole membrane and lumen, Ssh4 appears to be constitutively turned over, which will deplete the pre-existing Ssh4 pool.

*4) The strongest line of evidence against the direct invagination hypothesis in Li et al. Molecular Cell 2015 is the fact that upon lysine withdraw, Ypq1 was observed on structures detached from the vacuole, including: Ypq1-GFP punctae detached from the vacuole surface (Figure 6 and Figure S5B in Li et al.), Ypq1-nGFP and Vps23-cGFP split-GFP punctae detached from the vacuole surface (Figure S4 in Li et al.), and Ypq1-nGFP and cGFP-Ub split-GFP punctae detached from the vacuole surface (Figure S6D in Li et al.). The authors should discuss possible explanations of these earlier observations. Is it possible that Ypq1-SBP internalization and RapiDeg-induced Ypq1 internalization mainly go through the direct invagination route, whereas lysine withdrawal-induced Ypq1 internalization at least partially goes through an intermediate compartment?*

We have addressed these experiments in the Discussion. While we cannot rule out that some Ypq1 can be sorted to an intermediate compartment, we have shown that ubiquitinated Ypq1 readily sorts into the vacuole lumen in mutants that block vacuole delivery (*sec18ts, vam7ts, vps18ts*). We believe our data demonstrate that the trafficking of Ypq1-GFP, Ypq1-SBP and Ypq1 in the RapiDeg system are all mediated by the same pathway, since the recognition of the ub-cargo by ESCRTs can occur directly at the vacuole membrane as both signals for ESCRT binding — ub and PI3P are present on the vacuole.

*5) The authors show persuasively that brief shift to nonpermissive temperature for vps18 and vam3 tsf mutants prevents Ypq1 ubiquitination upon lysine withdrawal. They argue (Results section) that this is due to the absence of Ssh4 on the vacuole, but that key inference is not tested by looking at Ssh4 localization. It should be.*

We have included imaging data that shows that Ssh4 accumulates in pre-vacuolar compartments in both *vam3ts* and *vps18ts* cells at non-permissive temperature (Figure 2—figure supplement 1). We do not say that Ssh4 is absent from the vacuole membrane in these conditions, but that the pre-existing pool is not sufficient for ubiquitination.

*6) As a control for the blocks imposed by docking and fusion tsf mutants, Mup1 traffic through the MVB to the vacuole is examined. This is a good control but it involves a heterotypic rather than homotypic vacuole fusion event. Adding to this concern, it's a bit surprising to see wt/class A non-fragmented vacuole morphologies in the vps18tsf and vam7 tsf mutants after 45' (Figure 3) or 75' (Figure 3—figure supplement 1) at nonpermissive temperature.*

We do not see significant vacuole fragmentation in our experiments since they are done in relatively short time periods. Signiant vacuole fragmentation is seen after 6hrs of inactivation of these mutants, indicating that they block homotypic vacuole fusion.

*7) Since rapamycin treatment shuts off TORC1 with a cascade of effects on protein synthesis degradation, and stress signaling, more controls are critical. Missing controls include: monitoring of substrate reporters with mutant rapamycin non-binding FKBP domain; absence of FRB-Ub3; and/or the use of rapamycin-insensitive TOR1 alleles.*

We have clarified that the RapiDeg strain is rapamycin resistant in the Results section. The Material and methods previously described the construction of this strain: “WT strains (SEY6210 or SEY6210.1) were made rapamycin-resistant by transforming a PCR fragment encoding the dominant *TOR1* mutant (S1972I) and selected for growth on YPD plates (supplemented with 1μg/ml rapamycin). *FPR1 (FKBP-12*) was knocked out using homologous recombination”. We have included a control experiment showing rapamycin treatment does not induce Ypq1-FKBP degradation in the absence of FRB-3xUb (Figure 2—figure supplement 2).